# High-Yield Preparation of American Oyster Defensin (AOD) via a Small and Acidic Fusion Tag and Its Functional Characterization

**DOI:** 10.3390/md22010008

**Published:** 2023-12-20

**Authors:** Qingyi Zhao, Na Yang, Xinxi Gu, Yuanyuan Li, Da Teng, Ya Hao, Haiqiang Lu, Ruoyu Mao, Jianhua Wang

**Affiliations:** 1Gene Engineering Laboratory, Feed Research Institute, Chinese Academy of Agricultural Sciences, Beijing 100081, China; 2Innovative Team of Antimicrobial Peptides and Alternatives to Antibiotics, Feed Research Institute, Chinese Academy of Agricultural Sciences, Beijing 100081, China; 3Key Laboratory of Feed Biotechnology, Ministry of Agriculture and Rural Affairs, Beijing 100081, China; 4Enzyme Engineering Laboratory, College of Food Science and Technology, Hebei Agricultural University, Baoding 071001, China

**Keywords:** antimicrobial peptides, American Oyster Defensin (AOD), small fusion tag, pharmacodynamics, antimicrobial mechanism

## Abstract

The marine peptide, American oyster defensin (AOD), is derived from *Crassostrea virginica* and exhibits a potent bactericidal effect. However, recombinant preparation has not been achieved due to the high charge and hydrophobicity. Although the traditional fusion tags such as Trx and SUMO shield the effects of target peptides on the host, their large molecular weight (12–20 kDa) leads to the yields lower than 20% of the fusion protein. In this study, a short and acidic fusion tag was employed with a compact structure of only 1 kDa. Following 72 h of induction in a 5 L fermenter, the supernatant exhibited a total protein concentration of 587 mg/L. The recombinant AOD was subsequently purified through affinity chromatography and enterokinase cleavage, resulting in the final yield of 216 mg/L and a purity exceeding 93%. The minimum inhibitory concentrations (MICs) of AOD against *Staphylococcus aureus*, *Staphylococcus epidermidis*, and *Streptococcus galactis* ranged from 4 to 8 μg/mL. Moreover, time-killing curves indicated that AOD achieved a bactericidal rate of 99.9% against the clinical strain *S. epidermidis* G-81 within 0.5 h at concentrations of 2× and 4× MIC. Additionally, the activity of AOD was unchanged after treatment with artificial gastric fluid and intestinal fluid for 4 h. Biocompatibility testing demonstrated that AOD, at a concentration of 128 μg/mL, exhibited a hemolysis rate of less than 0.5% and a cell survival rate of over 83%. Furthermore, AOD’s in vivo therapeutic efficacy against mouse subcutaneous abscess revealed its capability to restrain bacterial proliferation and reduce bacterial load, surpassing that of antibiotic lincomycin. These findings indicate AOD’s potential for clinical usage.

## 1. Introduction

Antimicrobial peptides are a kind of cationic amphiphilic small molecule polypeptides that can be found in bacteria, fungi, animals, and plants [1]. These peptides possess a low molecular weight and are rich in basic amino acids such as lysine and arginine [2]. Antimicrobial peptides exhibit broad-spectrum antibacterial activity, in addition to having antifungal, antiviral, antiparasitic, and immunomodulatory functions [3,4,5]. The AOD peptide comprises thirty-eight amino acids and has three disulphide bond pairs (C4–C25, C11–C33, C15–C35). With a molecular weight of 4265 Da, it has an isoelectric point of 9.18 and carries a charge of +5. Additionally, hydrophobic amino acids make up approximately 34% of its composition [6]. It has been demonstrated that AOD possesses a potent inhibitory effect on *Escherichia coli*, *Staphylococcus aureus*, and other strains [6,7]. However, no occurrences of AOD biopreparation exist on account of its high charge and hydrophobicity.

In recent years, the co-expression technique involving fusion proteins has become widely popular for generating soluble recombinant proteins. Fusion proteins offer numerous advantages including increased expression levels, enhanced solubility, and convenient purification [8]. The commonly used fusion protein tags include His-tag [9], SUMO-tag [10], and GST-tag. The selection of the suitable tag depends on the molecular properties of the target, the utilized expression system, and the intended application of the end product. These tags can be removed enzymatically or chemically from the fusion protein. The Flag-tag, a fusion peptide made up of the acidic short sequence DYKDDDDK, is a short fusion tag that exhibits a recovery rate more than 10 times over than regular fusion labels (10–30 kDa) [11], resulting in a higher yield for the target product. With a charge number of −3, the Flag-tag provides efficient resistance to positive charges of antimicrobial peptides and facilitates the generation of stable fusion sequences that are easily expressed. The C-terminal of the Flag-tag comprises an enterokinase cleavage site (DDDDK) [12], which neither influences the biological activity nor the folding of the target protein. This enables the extraction of non-fused proteins in their natural form [13].

*Staphylococcus epidermidis* is a gram-positive bacteria that colonizes the skin, particularly on medical devices and in livestock farming, causing cutaneous infections or acute dermatological diseases in animals [14]. Because of its propensity for biofilm formation, *S. epidermidis* can become encapsulated within an extracellular matrix, resulting in chronic and recurrent infections even after antibiotic withdrawal, thus enabling pathogenic bacterial reinfection [15].

In this study, Flag-tags were used to construct recombinant expression vectors. The purified AOD products were obtained by enterokinase cleavage, and some druggability properties were characterized. In addition, a mouse model of *S. epidermidis* infection was established to observe the effect of AOD against *S. epidermidis* in vivo.

## 2. Results

### 2.1. Construction of the Recombinant Plasmids

The 6His-2Flag-AOD gene was subjected to enzymatic digestion using *Xho*I and *Xba*I, followed by insertion into plasmid pPICZαA and transformation into *Escherichia coli* DH5αH, resulting in the generation of the recombinant vector pPIC-6His-2Flag-AOD (Figure 1A). Subsequently, the *Pme*I was employed for linearizing the recombinant vector and transferring it to *Pichia pastoris* X-33. Positive transformants were selected using 100 μg/mL Zeocin.

### 2.2. Expression of 6His-2Flag-AOD at Shaker and Fermenter Level with High Level

The selected positive inverters were expressed in a shaker as shown in Figure 1B. After 72 h of induction, the total protein expression level reached approximately 80–100 μg/mL. The transformants with the highest yield were selected for high-density fermentation in a 5 L fermenter to increase the yield of fusion protein. The expression levels of the target protein increased gradually, reaching their peak at 72 h of induction. The total concentration and biomass reached 587 mg/L and 340 g/L, respectively (Figure 1C). 

### 2.3. Purification, Cleavage, and Identification

The 6His-2Flag-AOD protein was purified using a Ni-containing affinity column and detected by Tricine-SDS-PAGE (Figure 1D). Lane 3 at the penetration peak and lane 4 at the elution peak containing 25 mM imidazole showed no detection of the target protein. Meanwhile, lane 5 showed the corresponding elution peak of 6His-2Flag-AOD in the presence of 500 mM imidazole. The variation in imidazole content could be the possible cause of the 6His-2Flag-AOD’s diverse migration degrees seen in distinct SDS-PAGE diagrams. Subsequently, the enterokinase cleaved and purified product is shown in Figure 1E. Following cleavage, complete separation between the target protein and its attached label was achieved, resulting in the final yield of 216 mg/L and a purity of over 93%. Furthermore, MALDI-TOF MS analysis revealed that the purified AOD contained only one target peak of 4263.059 Da (Figure 1F), which was consistent with the theoretical molecular mass of 4265 Da.

### 2.4. AOD Had High Antibacterial Activity

The minimum inhibitory concentrations (MICs) of AOD against nine strains of gram-positive bacteria were determined, including *S. aureus* ATCC 43300, *S. aureus* ATCC 25923, *S. aureus* E48, *S. pseudintermedius* A2101, *S. agalactiae* ATCC 13813, *S. hyicus* NCTC 10350, and *S. epidermidis* G-81 (Table 1). It was demonstrated that AOD exhibited potent antibacterial activity against gram-positive bacteria such as *S. aureus* and *S. epidermidis* with low MICs ranging from 4 to 16 μg/mL. However, there was no bactericidal effect on *E. coli* ATCC 25922, *Salmonella typhimurium* CVCC 14028, and *Shigella flexneri* CMCC 51571, as their MICs were greater than 64 μg/mL.

### 2.5. AOD Eradicated S. epidermidis Thoroughly

The time-killing curves showed that the number (Log_10_ CFU/mL) of *S. epidermidis* G-81 decreased significantly after the AOD treatment (Figure 2A). After the administration of lincomycin, a gradual decrease in *S. epidermidis* G-81 was observed within 24 h. In the AOD treatment group, complete eradication of *S. epidermidis* G-81 occurred within 30 min and 1 h after 4× and 2× MIC treatments, and no resurgence was observed within 24 h. However, 1× MIC treatments showed a slight increase in bacterial counts at 2 and 4 h, indicating that these concentrations were unable to completely eliminate the bacteria within the initial 30 min. It was concluded that AOD has a dose-dependent bactericidal effect on target bacteria and has obvious advantages over lincomycin, which requires more time to thoroughly kill bacteria over 24 h. 

### 2.6. AOD Showed Low Hemolytic Activity and Cytotoxicity

The hemolytic activity of AOD was assessed by its impact on the integrity of mouse erythrocytes. As depicted in Figure 2B, the hemolysis caused by AOD at a concentration of 256 μg/mL was 1.29%, indicating that the peptide had a very low hemolytic potential. Additionally, as shown in Figure 2C, the cell survival rate at 128 mg/mL of AOD was 98%, indicating that AOD was not cytotoxic to Hacat cells. These results collectively support the non-toxic characteristic of AOD and its potential as a drug candidate for the treatment of bacterial infections. 

### 2.7. AOD Exhibited High Stability

The stability of AOD was assessed in terms of its response to protease, temperature, ions, and pH. It was demonstrated that AOD was tolerant to pepsin and trypsin (Figure 2D,E). The MIC values of AOD remained unchanged after treatment with pepsin and trypsin for 4 h. Additionally, excellent stability was observed for AOD at temperatures of 20, 40, and 80 °C. Meanwhile, its antibacterial activity decreased to 20% at the temperature of 100 °C (Figure 2F). AOD also showed remarkable stability when exposed to Zn^2+^, Mg^2+^, Ca^2+^, NH^4+^, K^+^, and Na^+^ (Figure 2G). Moreover, AOD displayed good tolerance to pH changes with no significant alteration in MICs within the range of pH 2.0–10.0 (Figure 2H). All the above results suggest that AOD has excellent stability.

### 2.8. AOD Disrupted the Bacterial Membrane Integrity of S. epidermidis (Flow Cytometry)

The impact of AOD on the integrity of the bacterial cell membrane is illustrated in Figure 3. In the absence of the AOD treatment, only 4.1% of *S. epidermidis* G-81 showed PI staining (Figure 3A), indicating that the cell wall and membrane remained relatively intact. After treatments at 1× MIC, 2× MIC, and 4× MIC of AOD, the proportions of PI-permeated bacteria were observed to be 34.6%, 44.5%, and 48.9%, respectively (Figure 3B–D). These results suggest that AOD induces dose-dependent damage on the cell membrane of *S. epidermidis* G-81. 

### 2.9. AOD Disrupted the Bacterial Membrane Integrity of S. epidermidis (SEM and TEM)

The morphological and microstructural changes of AOD-treated *S. epidermidis* G-81 were observed by SEM (Figure 4). As shown in Figure 4, untreated *S. epidermidis* G-81 exhibited a smooth surface and intact cellular morphology. After treatment with 4× MIC for 1 h, the surface of *S. epidermidis* G-81 became wrinkled, the cells were fragmented, and their contents leaked. 

The ultrastructural changes of AOD treated *S. epidermidis* G-81 were observed by TEM (Figure 5). In the control group without the AOD treatment, the cell membrane of *S. epidermidis* G-81 was intact and the electron density of the cytoplasmic contents was evenly distributed. After 4× MIC AOD treatment, the cell wall of *S. epidermidis* G-81 was broken, the contents leaked out, and the cell division at the division stage was uneven. 

### 2.10. AOD Showed Protective Effect in the Murine Abscess Model for S. epidermidis Infection

Infected mice showed a tendency to aggregate into clusters within one to four days post-infection, accompanied by a decrease in overall locomotor activity. There was no swelling at the inoculation site 24 h after infection. Three days later, a purulent discharge appeared with prominent lumps and ulceration. On the fourth day after infection, the abscess volume in the treated group was smaller than that in the infection group, and by the ninth day after infection, there was a significant reduction in abscess volume in the treated group. The volume of the abscess in the AOD group was smaller than that in the infection group. At the end of the experimental observation period, mice treated with 7.5 mg/kg AOD exhibited satisfactory recovery with no evident pathological swelling (Figure 6). Compared to the control group, infected mice experienced a significant weight loss. Meanwhile, the drug-treated group began to regain weight on the sixth day, whereas it took until the ninth day for the infection group to show weight gain (Figure 7A). Overall, the recovery rate of the infected group was slower than that of the treated group.

The AOD treatment significantly reduced the number of bacterias in the skin. The bacterial load in the skin of the treatment group was reduced by over 99% compared to the infection group on the seventh and fifteenth day post-infection (Figure 7B). The abscess in the AOD-treated mice completely disappeared at 15 days post-infection, indicating that AOD had superior efficacy compared to lincomycin (Figure 7C). The number of colonies per 100 mg in the organs of mice from each treatment group remained in the single digits, with minimal presence observed in the blood. Comparable to the treatment group, mice in the infected group had similar bacterial counts on day three; however, on days seven and fifteen, there was a significant overflow of bacteria in their bloodstream, indicating systemic infection. On day three post-infection, liver, spleen, and kidney indices were elevated compared to the control group while lung indices decreased. Interestingly, heart indices were elevated only in the AOD-treated group, in contrast to the other groups (Figure 7D–H).

## 3. Discussion

The AOD peptide, derived from North American mussels, is believed to be constitutively expressed as it is found in oyster unaffected by microorganisms. The C4 columns was employed for purification by preparative continuous AU-PAGE and reversed-phase HPLC [6]. It has a broad antibacterial spectrum and shows relatively high activity against gram-positive bacteria [6]. However, due to the very low yield from the extract of North American mussels, there is an urgent need for a high-level expression system to enable large-scale production of AOD. However, the high positive charge of AOD poses significant challenges for heterologous expression.

The Flag-tag is an eight amino acids fusion tag located at the N-terminal of the fusion, which can result in a higher theoretical yield compared to larger macromolecular fusion tags such as GST, SUMO, and Trx. Additionally, its −3 charge can effectively neutralize the positive charge of antimicrobial peptides, facilitating expression of the fusion sequence. At the C-terminal end of the Flag-tag, an enterokinase cleavage site is designed to allow complete separation of the tag from the target protein [11]. Enterokinase is an enzyme found in the intestine of animals that has a specific cleavage action on the trypsinogen [16]. Henceforth, the fusion protein is a plausible option for direct oral administration since the fusion protein can be self-cut and the fusion label will be removed orthotopically. The fusion protein 6His-2Flag-AOD was constructed by linking the Flag-tag with the antimicrobial peptide and the affinity label His. The AOD has a net charge of +5, whereas each Flag-tag bears a charge of −3. By incorporating two Flag-tags, the positive charge of the AOD can be efficiently neutralized, resulting in a fusion protein that is more stable. 

Initially, *E. coli* were employed as expression hosts, but the supernatant of the lysate following IPTG induction had almost no target protein detected. Additionally, a very low yield of target fusion protein (20 mg/L) was found in the precipitates, which contained inclusion bodies and cell fragments. In contrast, *P. pastoris* demonstrated a higher yield for the Flag-AOD expression and was subsequently chosen as the expression host. The results demonstrate that the total protein expression level reached 80–100 μg/mL following a 72 h incubation period in shaker culture. The expression of recombinant protein may increase by 5–10 folds when moving from a low-density shaker to a high-density fermenter [17,18,19]. Consequently, the total protein concentration of the fermentation supernatant reached 587 mg/L following the induction in a 5 L fermenter for 72 h (Figure 1). After purification via affinity chromatography and enterokinase dissection, the antimicrobial peptide displayed the final yield of 261 mg/L with a purity of over 93%.

The native AOD purified from American oyster displays a broad antibacterial spectrum against both gram-positive and negative bacteria with the minimal effective concentration (MECs) of 10 and 32 μg/mL to *S. aureus* and *E. coli*, respectively [16]. In this work, the AOD has strong bactericidal activity against gram-positive bacteria such as *S. epidermidis*, *S. aureus*, and *S. hyicus* with the MICs of 4–16 μg/mL (Table 1). However, there was no antimicrobial activity against gram-negative bacteria. This contradiction may be due to the differences in measurement methods between MEC [16] and MIC [20], and potential conformational changes in the spatial structure that may affect the activity. Hence, further research is necessary.

The time-bactericidal curve serves as an indicator to assess the druggability of candidates. Some antimicrobial agents including NZ2114 [20], PN7 [21] and Lysostaphin [22] have proven to be highly effective against most pathogens within a short period of time, without any subsequent bacterial regrowth. Consistent with these findings, our study demonstrated that *S. epidermidis* G-81 could be completely eradicated within one hour at 2× and 4× MIC, and no bacterial regrowth was observed within 24 h (Figure 2A). Besides effectiveness, stability and toxicity are both essential factors that influence the practical application of medication. The biocompatibility of AOD has not been studied in previous studies [16,17]. The study demonstrates that AOD showed a hemolysis rate of less than 0.5% at 256 μg/mL, alongside cell survival rates exceeding 83% (Figure 2B,C). Moreover, the bactericidal activity of AOD remained unimpaired after four hours of exposure to artificial gastric and artificial intestinal juice, indicating its resistance to pepsin and trypsin and thus supporting its potential for oral administration.

The presence of particular salt ions (Na^+^, Mg^2+^ and Ca^2+^) can reduce the activity of some antimicrobial peptides, like FNZ [23] and PN7 [21]. Following a two-hour treatment with K^+^, Na^+^, Mg^2+^, Zn^2+^, Ca^2+^ and NH_4_^+^, the activity of AOD remained intact, indicating its excellent stability in different ion environments (Figure 2G). Thermal and pH stability are pivotal in the application of antimicrobial peptides [19]. Remarkably, AOD displays extraordinary thermal stability in a temperature range of 20–80 °C coupled with excellent resistance to a wide pH range from pH 2 to 10 (Figure 2H). These in vitro antimicrobial characteristics highlight the potential suitability of AOD as a pharmaceutical agent.

A major part of the bactericidal mechanism of most of the antimicrobial peptides is directly on bacterial cell membranes, which have negative charges [24]. Flow cytometry was employed to observe the damage effect of AOD on the cell membrane of *S. epidermidis* G-81. It is apparent that AOD has a higher damage effect on the cell membrane of bacteria, with the penetration rate of 4× MIC AOD being 48.9% (Figure 3). Additionally, after a one-hour exposure to 4× MIC AOD, the cell wall of *S. epidermidis* G-81 was disrupted, resulting in the leakage of cellular contents and irregular cell division during the division phase in the AOD-treated group. These findings are consistent with those obtained through SEM and TEM (Figure 4 and Figure 5). It has been demonstrated that AOD’s bactericidal mechanism involves structural disruption of the cell membrane, resulting in membrane rupture and pore formation, ultimately culminating in cell lysis. However, its interaction with liposomes or other macromolecules on the cell membrane needs to be further studied [25]. 

The development of a simulated clinical infection animal model is an essential pre-requisite for drugs evaluation. It was demonstrated that administering BP2 (5 m/kg) one hour after the *S. epidermidis* challenge in the biomaterial-associated infection model (BAI) led to an 80% reduction in the pathogens. Following a 24 h challenge, the survival rate of *S. epidermidi* in peri-implant tissue was reduced by 100 folds [26]. In a previous study, we set up a murine abscess model for *S. epidermidis* infection and evaluated the efficacy of NZX [27]. It was found that AOD exhibited superior anti-*S. epidermidis* activity in mice compared to lincomycin. Additionally, after administering AOD treatment to mice with subcutaneous abscess, the volume of abscess was significantly smaller, and the bacteria load of skin was reduced, indicating a degree of protective effect on organs. These findings suggest that AOD is a viable in vivo treatment and shows potential as a candidate for future drug development.

## 4. Materials and Methods

### 4.1. Strains, Plasmids, Reagents, Cell Line, and Animals

*Escherichia coli* DH5α, plasmid pPICZαA, and *P. pastoris* X-33 were purchased from Invitrogen (Beijing, China) and used for cloning and expression. The *S. aureus* ATCC 43300 and *S. aureus* ATCC 25923 were purchased from American Type Culture Collection (ATCC). *S. epidermidis* G-81 was a present of Professor Wu from China Agricultural University (Beijing, China). The kits for plasmid extraction and DNA purification were purchased from Tiangen Co., Ltd. (Beijing, China). The AOD nucleotide sequence was synthesized by Sangon Biotech Co., Ltd. (Shanghai, China). Hacat cells were purchased from Peking Union Medical College. The recombinant enterokinase with a specification of 1000 U and product number P4237–1000 U was acquired from Biotime Biotechnology Co., Ltd. (Beijing, China). Additionally, six-week-old specific-pathogen-free (SPF) female BALB/c mice (approximately 20 g/mouse) were purchased from the Vital River Laboratories (VRL, Beijing, China). Microbial and animal cell experiments were conducted within Class II biological safety cabinets. All other chemical reagents used were analytical grade.

### 4.2. Expression, Purification, Cleavage, and Identification of AOD

#### 4.2.1. Construction of the Recombinant Plasmids

The 6His-2Flag-AOD gene was optimized based on the codon preference of *P. pastoris* (www.kazusa.or.jp/codon/, accessed on 13 April 2022) by Reverse Translate Tool (www.bio-informatics.org/sms2/rev_trans.html, accessed on 16 April 2022). T4 DNA ligase was employed to connect DNA fragments to pPICZαA plasmid. The recombinant vector was linearized by *Pme*I and transformed into the susceptible *P. pastoris* X-33 through electroporation, and the pPICZαA vector was utilized as a negative control [20]. Positive transformants were screened on YPDS plates containing 100 μg/mL Zeocin (1% yeast extract, 2% glucose, 2% peptone, 2% agar, and 1 M sorbitol) and identified by colony PCR [28].

#### 4.2.2. Expression of 6His-2Flag-AOD at the Shaking Flask and Fermenter Level

Positive inverters were selected and cultured in YPD liquid medium supplemented with Zeocin at a concentration of 100 μg/mL, followed by overnight incubation at 30 °C for 14–18 h. Subsequently, the culture was transferred to 200 mL of BMGY medium. The 1% methanol was added every 24 h for a total of 120 h induction [20]. Finally, the supernatant fermentation solution was collected and subjected to verification by SDS-PAGE.

The 5 L level fermentation was performed according to previous protocols [29]. 

#### 4.2.3. Purification, Identification, and Cleavage of 6His-2Flag-AOD

The fermentation solution was centrifuged at 5000 rpm for 30 min and the supernatant was collected. The 6His-2Flag-AOD was purified by ion exchange column as the following: the ion exchange column was balanced with a 20 mM phosphate buffer (pH = 7.4). The elution buffer containing 25 mM imidazole (pH 5.7) was used to elute miscellaneous protein and the target protein was eluted using a 20 mM phosphate buffer containing 500 mM imidazole. After elution, SDS-PAGE analysis confirmed the target protein. The purified product of 6His-2Flag-AOD (0.1–1 mg/mL) was enzymatically cleaved in the 25 mM Tris-HCl buffer at 25 °C for 16 h with 0.1–0.2 U enterokinase. Subsequently, dialysis was employed to remove ions, and Tricine-SDS-PAGE analyses were used to verify the effectiveness of cleavage. Finally, an ultrafiltration tube with a molecular weight cutoff of 10 kD was utilized; it underwent centrifugation at 8000 rpm for 10 min to collect the lower liquid phase, which then underwent freeze-drying before storage. The lyophilized powder of AOD was identified by matrix-assisted laser desorption/ionization-time of flight mass spectrometry (MALDI-TOF MS).

### 4.3. Physical and Chemical Properties

#### 4.3.1. Minimal Inhibitory Concentrations

The MIC value of AOD was determined by microbroth dilution method [30]. The strains to be tested were cultured overnight at 37 °C and then transferred to fresh medium until the logarithmic phase of growth was reached. Bacteria were diluted to a concentration of 1 × 10^5^ CFU/mL and added into the 96-well plate, followed by the addition of a series of serially diluted AOD. The plate was then incubated at 37 °C for 12–18 h. The minimum concentration of no bacterial growth was recorded as the MIC value. Each experiment was performed in triplicate.

#### 4.3.2. Bactericidal Effect of AOD on S. epidermidis G-81 In Vitro

The time-killing curves were employed to evaluate the efficacy of AOD against *S. epidermidis* G-81 in vitro [31]. *S. epidermidis* G-81 was cultured overnight and subsequently transferred to fresh medium until the logarithmic phase of growth was reached. It was diluted to a concentration of 1 × 10^5^ CFU/mL, and was exposed to 1×, 2× and 4× MIC of AOD. Samples were collected at intervals of 0, 0.5, 1, 2, 4, 6, 8, 10, 12, and 24 h, respectively, and subjected to 10-fold serial dilutions followed by colony counting. PBS and Lincomycin were used as negative and positive controls. Each experiment was performed in triplicate, and data were analyzed by the GraphPad Prism (version 8, USA, the software download from https://www.graphpad-prism.cn/, accessed on 7 June 2023). The results were presented as means ± standard deviation (SD).

#### 4.3.3. Cytotoxicity, Hemolysis, and Stability of AOD

##### Hemolysis

Erythrocytes from mice of SPF grade were centrifuged at 2000 rpm for 5 min, followed by washes with saline solution three times. Subsequently, the 100 μL of 8% erythrocytes (diluted in 0.9% NaCl) was combined with an equal volume of serially diluted AOD solution. The mixture was then incubated at 37 °C for 1 h. Subsequently, it was centrifuged at 5000 rpm for 5 min and the supernatant was collected for measurement of the absorbance at OD540 nm. The 0.9% NaCl and 0.1% Triton X-100 were used as blank (A0) and positive (A100) controls, respectively [32]. The percentage of peptide hemolysis is calculated as follows: Hemolysis (%) = [(A − A0)/(A100 − A0)] × 100. Each experiment was performed in triplicate, and data were analyzed by the GraphPad Prism (version 8, USA, the software download from https://www.graphpad-prism.cn/, accessed on 7 June 2023). The results were presented as means ± standard deviation (SD).

##### Temperature, pH, Ion, and Protease Stability

The thermal stability of AOD was determined by incubating it at temperatures of 20, 40, 60, 80, 100 °C for one hour in PBS [18]. The pH stability of AOD was assessed by incubating it in various buffers including glycine-HCl buffer (pH 2.0), sodium acetate buffer (pH 4.0), sodium phosphate buffer (pH 6.0), tris-HCl buffer (pH 8.0), and Glycine-NaOH buffer (pH 10.0). To evaluate the stability in the presence of proteases, AOD was mixed with pepsin (3000 U/mg, pH 2.0) and trypsin (250 U/mg, pH 8.0) at a ratio of 10:1 and incubated at 37 °C for 4 h. AOD was also incubated with 150 mM sodium chloride, 4.5 mM potassium chloride, 6 μM ammonium chloride, 8 μM zinc chloride, 1.25 mM calcium chloride, 1 mM magnesium chloride, and 4 μM ferric chloride at 37 °C for 4 h to test ion stability. In all aforementioned assays, the untreated AOD served as the positive control while the buffer alone acted as the negative control. The minimum inhibitory concentration was used to determine the relative antibacterial activity of AOD to *S. epidermidis*. Each experiment was performed in triplicate, and data were analyzed by the GraphPad Prism (version 8, USA, the software download from https://www.graphpad-prism.cn/, accessed on 7 June 2023). The results were presented as means ± standard deviation (SD).

##### Cytotoxicity

The cytotoxicity of AOD was assessed by using the WST-8 (2-(2-methoxy-4-nitrophenyl)-3-(4-nitrophenyl)-5-(2,4-disulfonylbenzene)-2h-tetrazole monosodium salt (cck-8) assay [33]. The Hacat cells were cultured in 96-well plates at a density of 2.5 × 10^5^ cells/well and incubated for 24 h. AOD was serially diluted (1–128 μg/mL) and added to each well, followed by a 24 h incubation period. After discarding the supernatant, the wells were washed twice with PBS, and a diluted cck-8 was added for a 2 h incubation. Absorbance was measured at OD570 nm and recorded as “A”. The absorbance of PBS was used as “A control”. Cell survival rate is calculated as follows: Survival rate (%) = (A/Acontrol) × 100. Each experiment was performed in triplicate, and data were analyzed by the GraphPad Prism (version 8, USA, the software download from https://www.graphpad-prism.cn/, accessed on 7 June 2023). The results were presented as means ± standard deviation (SD).

### 4.4. In Vitro Antibacterial Mechanism of AOD

#### 4.4.1. Membrane Permeabilization Analysis by Flow Cytometer

Flow cytometry was used to assess the impact of AOD on the integrity of *S. epidermidis* G-81 cell membrane. The *S. epidermidis* G-81 was cultured overnight at 37 °C and then transferred to fresh medium to reach the logarithmic phase of growth. The samples were subsequently washed three times with PBS and diluted to a density of 1 × 10^9^ CFU/mL. They were then incubated with AOD at 37 °C for 30 min. Afterward, the mixture was washed twice with PBS. Prior to flow cytometric analysis, the PI dye was added to a final concentration of 50 μg/mL and incubated for 15 min at room temperature. The negative control was 0.1 M PBS solution [34]. Data were analyzed by the software CellQuest Pro (version 5.1).

#### 4.4.2. Scanning Electron Microscopy Observations

The *S. epidermidis* G-81 strain was cultured overnight and then transferred to the logarithmic growth phase. Following that, it was diluted with PBS to a concentration of 1 × 10^9^ CFU/mL and incubated with 4× MIC AOD at 37 °C for 1 h. The resulting mixture was washed three times with PBS and fixed overnight at 4 °C with a solution containing 2.5% glutaraldehyde. After fixation, the supernatant was discarded, followed by dehydration through sequential immersion in ethanol solutions ranging from fifty percent concentration to three times absolute ethanol, each immersion lasting fifteen minutes per step. Finally, critical point drying and platinum sputtering were carried out prior to observation on the QUANTA200 SEM (FEI, Philips, The Netherlands) [32].

#### 4.4.3. Transmission Electron Microscopy Observations

The *S. epidermidis* G-81 strain was cultured until it reached the logarithmic growth phase. It was then diluted with PBS to obtain a bacterial suspension with a concentration of 1 × 10^8^ CFU/mL. AOD was added with a final concentration of 4× MIC, followed by incubation at 37 °C for 2 h. After incubation, the sample was centrifuged at 4000 rpm for 5 min at room temperature, followed by three washes with PBS and fixation in a 2.5% glutaraldehyde solution overnight at 4 °C. After being washed five times with PBS for seven minutes each, the bacteria were fixed in a solution containing 1% osmic acid for an hour. Subsequently, the bacteria underwent three more PBS washes, and dehydration was conducted using 1 mL acetone via a series of sequential steps using progressively increasing concentrations (50–70–85–95–100%) after rinsing. The bacteria were immersed in a mixture of ethanol and embedding solution (1:1) for two hours before being immersed in fresh embedding solution overnight. This process was repeated after changing the solution. The samples were then treated in an oven at 60 °C for twenty-four hours and stained using a 1% uranoxy acetate solution. The stained samples were sliced and observed under a transmission electron microscope [5]. 

### 4.5. Efficacy of AOD against S. epidermidis G-81 In Vivo

The *S. epidermidis* G-81 strain was cultured in TSB medium overnight and transferred to fresh medium until the logarithmic phase of growth was reached. The sterile saline was used for rapid washing twice.

The six-week-old BALB/c mice were divided randomly into five groups of nine animals: a normal saline control group, an infection group, three drug treatment groups (including AOD treatment in two groups and lincomycin treatment in one group). The hair removal solution was applied to the abdomen of each mouse in every group. One day later, each group, except for the control group, was subcutaneously injected with 0.1 mL of *S. epidermidis* G-81 at a concentration of 1 × 10^10^ CFU/mL. After 24 h, AOD (5 and 7.5 mg/kg, body weight) and lincomycin (7.5 mg/kg, body weight) were, respectively, injected into the infected subcutaneous site. In contrast, the infected group received an injection of 10 μL normal saline. The treatment duration was three days, and we continuously observed the weight and volume changes in the abscesses for 15 days. On days 4, 9, and 14 post-infection, three mice from each group were subjected to histopathological [35], organ index, and etiological examinations [36,37]. Each experiment was performed in triplicate, and data were analyzed by the GraphPad Prism (version 8, USA, the software download from https://www.graphpad-prism.cn/, accessed on 7 June 2023). The results were presented as means ± standard deviation (SD).

## 5. Conclusions

In summary, the Flag-tag was utilized as the fusion partner to develop a high-yield method for preparing the antimicrobial peptide AOD, resulting in a high expression level of 587 mg/L after 72 h of induction. Additionally, AOD displays notable antibacterial and pharmacological activity by effectively exerting its bactericidal effect against *S. epidermidis* G-81 through the destruction of both cell wall and membrane. In vivo experiments further confirmed the therapeutic effectiveness of AOD in the treatment of mouse abscesses, demonstrating its potential as a novel bactericidal agent with promising clinical usefulness.

## Figures and Tables

**Figure 1 marinedrugs-22-00008-f001:**
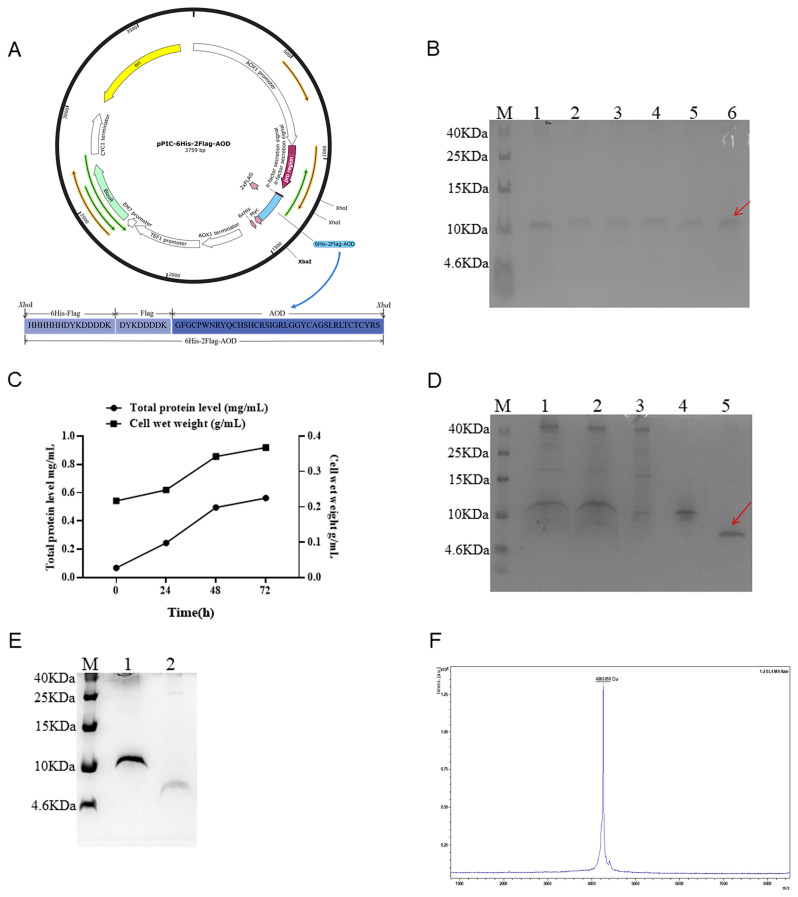
Expression, purification, and cleaving of 6His-2Flag-AOD and the identification of AOD. (**A**) Schematic diagram of recombinant expression plasmid pPIC6His-2Flag-AOD. (**B**) The expression level of 6His-2Flag-AOD in shaker was analyzed by Tricine-SDSPAGE. Lane M represents protein molecular weight marker (7 µL). Lanes 1–6 represent different positive converters. The fusion protein 6His-2Flag-AOD is represented by the red arrow. (**C**) Time curves of total secreted protein levels and cell wet weight at 0, 24, 48, and 72 h induced by high-density fermentation. (**D**) Tricine-SDS-PAGE analysis of collected eluents purified by affinity chromatography. Lane M represents protein molecular weight marker (7 µL). Lane 1 10 µL unpurified fermentation supernatant; Lane 2 filters the unpurified fermentation supernatant after filtration with a 0.45 µm filter; Lane 3, material unretained by the column; Lane 4 eluent collected at 25 mM imidazole concentration; Lane 5 collects 10 µL eluent at the target peak (red arrow). (**E**) AOD after cleavage with enterokinase. Lane M represents protein molecular weight marker (7 µL). Lane 1 uncut 6His-2Flag-AOD; Lane 2 AOD after cutting. (**F**) MALDI-TOF MS analysis of the purified AOD.

**Figure 2 marinedrugs-22-00008-f002:**
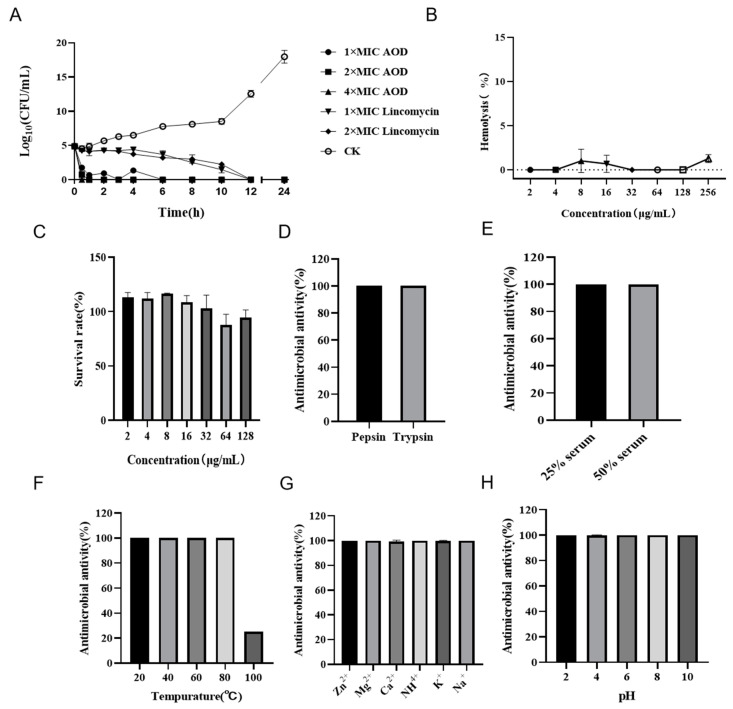
Bactericidal curve, hemolysis, cytotoxicity, and stability of AOD. (**A**) Time-killing curve of AOD (1×, 2×, or 4× MIC) against *S. epidermidis* G-81. Lincomycin (1× and 2× MIC) and PBS were used as positive and negative controls, respectively. “CK” is the PBS-treated negative control. (**B**) The hemolytic activity of AOD at different concentrations (1–256 μg/mL) to mouse red blood cells. (**C**) The cytotoxicity of AOD at different concentrations (1–128 μg/mL) to Hacat cells. (**D**–**H**) The stability of AOD to protease, serum, temperature, ions, and pH.

**Figure 3 marinedrugs-22-00008-f003:**
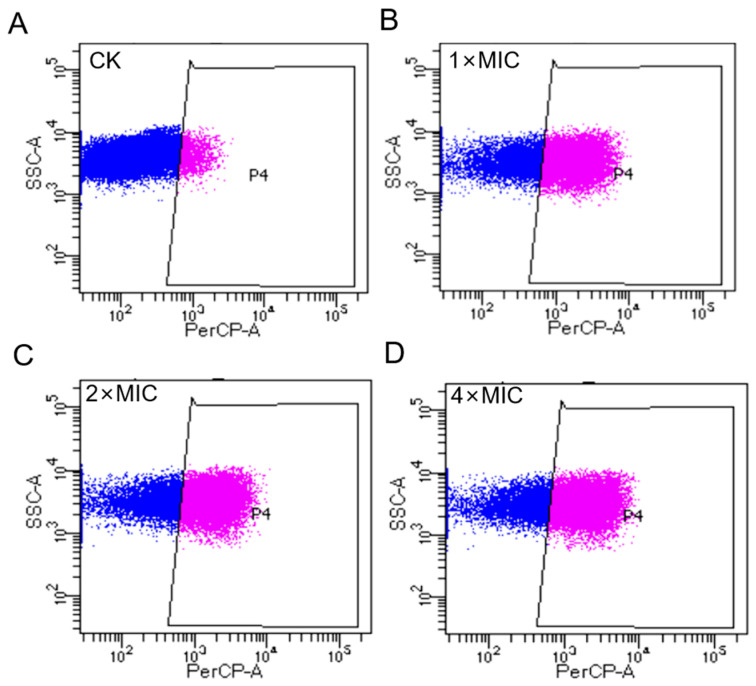
Flow cytometry analysis of the effect of AOD on cell membrane to *S. epidermidis* G-81. (**A**) The *S. epidermidis* G-81 without the AOD treatment (CK). (**B**–**D**) were flow cytograms after 1×, 2×, and 4× MIC AOD treatment.

**Figure 4 marinedrugs-22-00008-f004:**
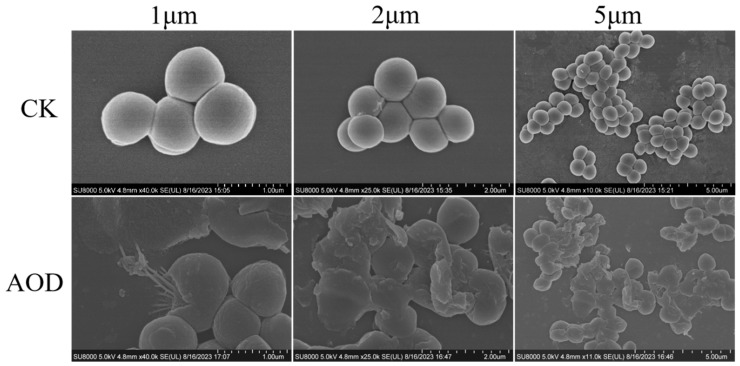
Scanning electron microscopy analysis of impact of AOD on *S. epidermid* G-81. After 4× MIC AOD treatment for 1 h, the samples were fixed, dehydrated, sliced, and subsequently observed via SEM. “CK” is the PBS-treated negative control.

**Figure 5 marinedrugs-22-00008-f005:**
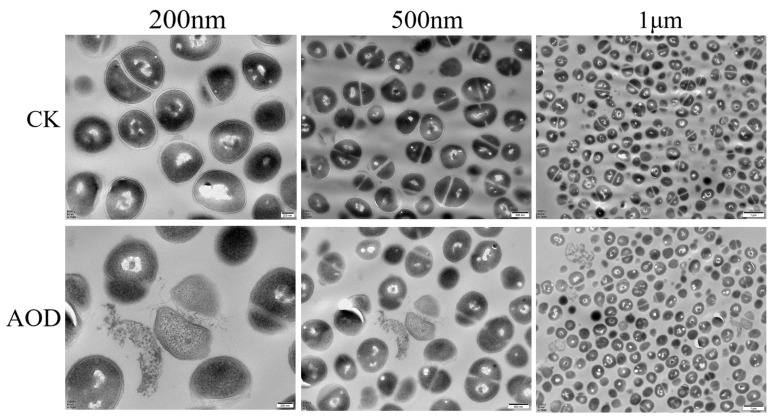
Transmission electron microscopy analysis of the effect on *S. epidermidis* of AOD at 4× MIC after treatment for 1 h. “CK” is the PBS-treated negative control.

**Figure 6 marinedrugs-22-00008-f006:**
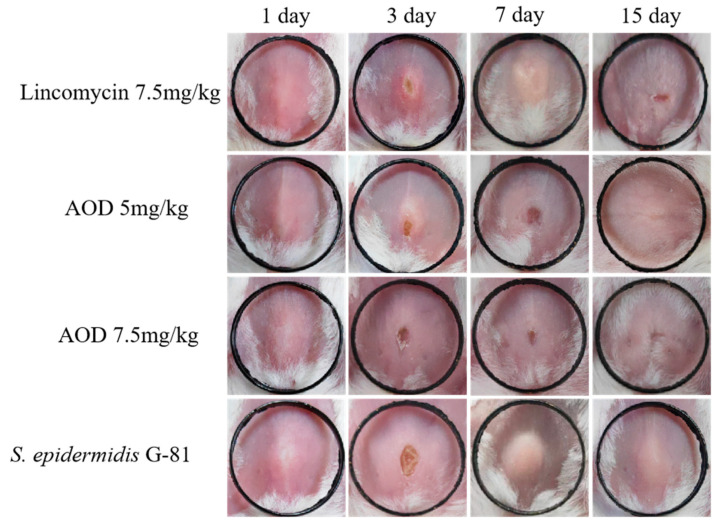
Therapeutic effect of AOD on a mouse model of skin abscess induced by *S. epidermidis* G-81. Photographs of abscesses of mice treated with AOD and lincomycin 24 h after subcutaneous injection of in the fourth raw CK, with black rings of 1 cm in length and width. “CK” is the PBS-treated negative control.

**Figure 7 marinedrugs-22-00008-f007:**
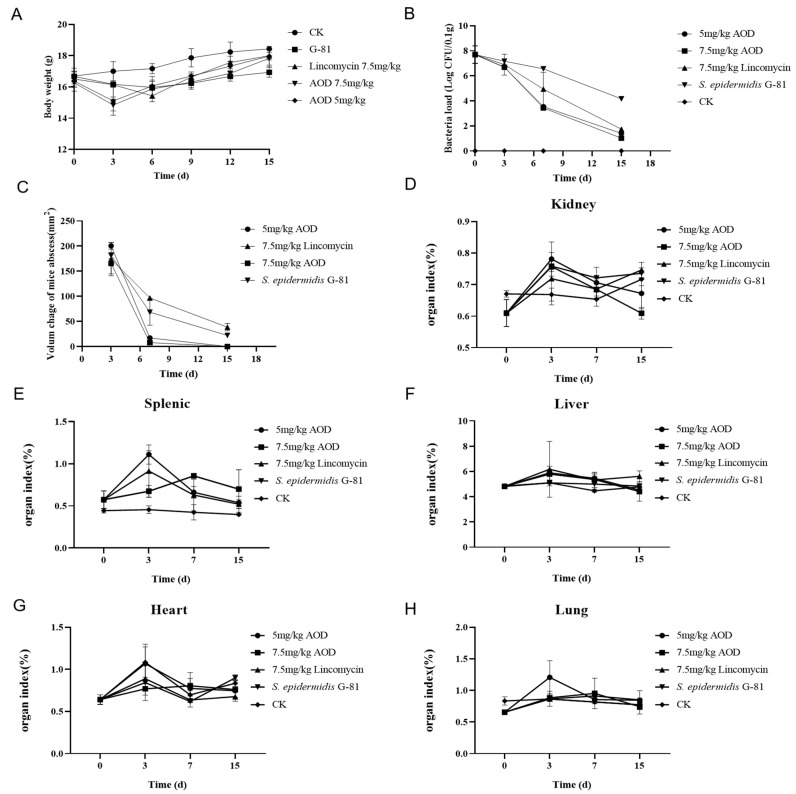
Therapeutic efficacy of AOD in a murine model of *S. epidermidis* G-81-induced skin abscess. (**A**) Assessment of body weight changes in mice; (**B**) Quantification of bacterial load in mouse skin tissue per 0.1 g; (**C**) Evaluation of the volume dynamics of mouse abscesses; (**D**–**H**) represent alterations in organ indices.

**Table 1 marinedrugs-22-00008-t001:** The MIC values of AOD and lincomycin.

Strains	MIC (μg/mL)	
	AOD	Lincomycin
*S. aureus* ATCC 43300	8	>64
*S. aureus* ATCC 25923	16	1
*S. aureus* E48	8	1
*S. pseudintermedius* A2101	8	32
*S. agalactiae* ATCC 13813	4	-
*S. hyicus* NCTC 10350	16	32
*S. epidermidis* ATCC 35984	4	-
*S. epidermidis* ATCC 12228	4	4
*S. epidermidis* G-81	4	4
*E. coli* ATCC 25922	>64	2
*Salmonella typhimurium* CVCC 14028	>64	-
*Shigella flexneri* CMCC 51571	>64	-

-: No tested.

## Data Availability

The original contributions presented in the study are included in the article; further inquiries can be directed to the corresponding author(s).

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
