# Peer review of "High-Yield Preparation of American Oyster Defensin (AOD) via a Small and Acidic Fusion Tag and Its Functional Characterization"

_marinedrugs, 2023, doi:10.3390/md22010008_

Round 1

Reviewer 1 Report

Comments and Suggestions for Authors

Zao et al., in their paper “High yield preparation of American oyster defensin (AOD) via a novel and small fusion tag and its functional characterization” describe the recombinant production of the American oyster defensin (AOD) in Pichia pastoris. After purification they characterize the antimicrobial activity of AOD, as MIC values and Time killing curves, as well as the resistance of peptide to various physico-chemical conditions, and its cytotoxicity. In addition, they investigate the effect of the peptide on the morphology of S. epidermidis using SEM and TEM, and its therapeutic efficacy on a mouse model of skin infection developed after subcutaneous injection of S. epidermidis.

The title in the part “High yield preparation …” is misleading as the final yield of AOD peptide is missing in the paper and so it cannot be inferred if the yield is high (see also below comment to lines 20 and 88).

The title in the part “… via novel and small fusion tag…” is misleading as the tag is not novel and is simply a combination of the well-known tags (His-tag and Flag-tag), which has already been used and in fact is cited by the authors themselves (ref. 11). (see also below comment tolines 18 and 69-70).

The manuscript is not particularly well written, the use of English requires reviewing and is at times difficult to follow.

There follows a list of comments:

Lines 20, 88 and 459-460: The authors indicate that the total protein concentration in the supernatant of the expression system was 587 mg/L but do not indicate the fraction of correct AOD, so that one cannot have any idea of the final expression yield. The final amount of AOD with respect to total supernatant protein should thus be reported.

Fig.1D the SDS PAGE indicates the presence of contaminating material (lanes 1 and 2) and furthermore the size of protein varies somewhat in the different bands. Furthermore, the apparent size of both the tagged AOD and the final cleavage product seem higher than expected. Are the SDS PAGE run under reducing conditions to cleave S-S bonds? If so, this should be indicated. Fig 1B is rather indistinct and only one species can vaguely be discerned. Does this correspond to tagged AOC? Furthermore, the loading of the bands appears to be quite low, and unlikely to reveal the presence of contaminants. Fig1A and 1F text is tiny and illegible

Table 1: species names should be indicated correctly (eg. S. pseudointermedius with lowercase p, ecc.) and furthermore different strains are reported in the table and in the text (is CVCC546 at line 116 the same as E48 in the table? is the strain 437-2 at line 117 the same as NCTC 10350 in the table?)

Line 18: This concept is unclear (what does yield lower than 20 % mean?)

Line 36: “Antimicrobial peptides are a kind of amphiphilic small molecule polypeptides…” instead of “Antimicrobial peptide is a kind of amphiphilic small molecule polypeptide…”

Line 42: ….charge of +5 

Lines 123 and 124: epidermidis instead of epidermiodis

Line 134: hemolytic activity

Line 136: mouse not mice

Fig. 2: 2A) What is CK. 2F) What happens to the peptide at 100°C. 

Line 254: The paper by Hopp et al. A short polypeptide marker sequence useful for recombinant protein identification and purification. Bio-Technology 1988, 6,1204–1210 referring to the Flag tag should be cited

Line 298: What is the BAI model?

Line 323: ..and identification of AOD

Lines 344, 345, 346 and 348: concentrations are in mM not mm

Lines 356, 365 and 443: “ were cultivated overnight and then transferred to fresh medium until the logarithmic phase of growth was reached …” instead of “… then transferred to the logarithmic stage.”

Lines 359 and 374: serially diluted instead of gradually diluted

Line 373: NaCl concentration is 0.9%, not 9%; diluted to 8% (vol/vol)

Line 381: in which buffer was the thermal stability measured?

Lines 445-447: “… divided into four groups of nine animals each”  one saline control group, infection group, two groups treated with AOD and one with lincomycin, so the groups are five.

Comments on the Quality of English Language

The manuscript is not particularly well written, the use of English requires reviewing and is at times difficult to follow.

Author Response

Response to the comment of Reviewer 1# 

Question1:The title in the part “High yield preparation …” is misleading as the final yield of AOD peptide is missing in the paper and so it cannot be inferred if the yield is high (see also below comment to lines 20 and 88).

The title in the part “… via novel and small fusion tag…” is misleading as the tag is not novel and is simply a combination of the well-known tags (His-tag and Flag-tag), which has already been used and in fact is cited by the authors themselves (ref. 11). (see also below comment tolines 18 and 69-70).

Answer 1: Thank you for your comments. The title has been changed to “High yield preparation of American Oyster Defensin (AOD) via small and acidic fusion tag and its functional characterization”, and the final yield of AOD (216 mg/L) was added in the abstract and main text.

Question 2: The manuscript is not particularly well written, the use of English requires reviewing and is at times difficult to follow.

Answer 2: Thank you for your comments. The grammatical errors and colloquial expressions were revised. Additionally, the language has been improved by the Elsevier Language Editing Services (No. ASLESTD1247326).

Question 3: Lines 20, 88 and 459-460: The authors indicate that the total protein concentration in the supernatant of the expression system was 587 mg/L but do not indicate the fraction of correct AOD, so that one cannot have any idea of the final expression yield. The final amount of AOD with respect to total supernatant protein should thus be reported.

Answer3: Thank you for your suggestion. The final yield of AOD (216 mg/L) was added in the abstract and main text.

Question 4: Fig.1D the SDS PAGE indicates the presence of contaminating material (lanes 1 and 2) and furthermore the size of protein varies somewhat in the different bands. Furthermore, the apparent size of both the tagged AOD and the final cleavage product seem higher than expected. Are the SDS PAGE run under reducing conditions to cleave S-S bonds? If so, this should be indicated. Fig 1B is rather indistinct and only one species can vaguely be discerned. Does this correspond to tagged AOC? Furthermore, the loading of the bands appears to be quite low, and unlikely to reveal the presence of contaminants. Fig1A and 1F text is tiny and illegible

Answer 4: Thank you for your suggestions. 1) Lanes 1 and 2 in Fig.1D represent the fermentation supernatant and the fermentation supernatant after filtration by 0.45 μm filter, respectively, which have been described in line 109-111. 2) The presence of different buffers in the samples and standard marker may lead to a discrepancy between the electrophoretic bands and the expected molecular weight. Moreover, the results from mass spectrometry provide evidence that the AOD's molecular weight matches the expected value (Fig1 F). 3) B and F in figure 1 have been.

Question 5: Table 1: species names should be indicated correctly (eg. S. pseudointermedius with lowercase p, ecc.) and furthermore different strains are reported in the table and in the text (is CVCC546 at line 116 the same as E48 in the table? is the strain 437-2 at line 117 the same as NCTC 10350 in the table?)

Answer 5: Thank you for bringing this error. The species name has been revised and corrected accordingly. The typographical error has been rectified.

Question 6: Line 36: “Antimicrobial peptides are a kind of amphiphilic small molecule polypeptides…” instead of “Antimicrobial peptide is a kind of amphiphilic small molecule polypeptide…”

Answer 6: Thank you for your comments. It has been revised.

Question 7: Line 42: ….charge of +5

Answer 7: Thank you for your suggestion. It has been revised.

Question 8: Lines 123 and 124: epidermidis instead of epidermiodis

Answer 8: Thank you for your comments. It has been revised.

Question 9: Line 134: hemolytic activity

Answer 9: Thank you for your comments. It has been confirmed.

Question 10: Line 136: mouse not mice

Answer 10: Thank you for your comments. It has been revised.

Question 11: Fig. 2: 2A) What is CK. 2F) What happens to the peptide at 100°C.

Answer 11: Thank you for your comments. 2A) “CK” is the PBS-treated negative control, and it was added in the figure legend of Fig.2. 2F) The minimum inhibitory concentration (MIC) of AOD against Staphylococcus epidermidis increased from 4 to 16 μg/mL following exposure to 100℃ for an hour. AOD was deactivated under these conditions, with a mere 25% of its activity remaining.

Question 12: Line 254: The paper by Hopp et al. A short polypeptide marker sequence useful for recombinant protein identification and purification. Bio-Technology 1988, 6,1204–1210 referring to the Flag tag should be cited

Answer 12: Thank you for your suggestion. The reference has been cited.

Question 13: Line 298: What is the BAI model?

Answer 13: Thank you for your question. The BAI model is the “biomaterial-associated infection model”. The backs of mice were shaved and cleansed with 70% ethanol before making a 0.3 cm incision 1cm outside each spine. Then, a 1cm SEpvp segment was implanted subcutaneously using a transponder to minimize tissue damage.

Question 14: Line 323: ..and identification of AOD

Answer 14: Thank you for your comments. It has been revised.

Question 15: Lines 344, 345, 346 and 348: concentrations are in mM not mm

Answer 15: Thank you for your comments. It has been revised.

Question 16: Lines 356, 365 and 443: “ were cultivated overnight and then transferred to fresh medium until the logarithmic phase of growth was reached …” instead of “… then transferred to the logarithmic stage.”

Answer 16: Thank you for your suggestion. It has been revised.

Question 17: Lines 359 and 374: serially diluted instead of gradually diluted

Answer 17: Thank you for your suggestion. It has been revised.

Question 18: Line 373: NaCl concentration is 0.9%, not 9%; diluted to 8% (vol/vol)

Answer 18: Thank you for your comments. It has been revised to “with saline solution and a dilution to 8% (vol/vol)”.

Question 19: Line 381: in which buffer was the thermal stability measured?

Answer 19: Thank you for your comments. The peptide was dissolved in PBS buffer and incubated at various temperatures for two hours to ascertain the effectiveness of AOD.

Question 20: Lines 445-447: “… divided into four groups of nine animals each” one saline control group, infection group, two groups treated with AOD and one with lincomycin, so the groups are five.

Answer 20: Thank you for your comments. The groups are five and it has been revised.

Reviewer 2 Report

Comments and Suggestions for Authors

The study by Zhao et al. successfully established recombinant production of an antibacterial peptide, AOD. In addition, as their recombinant AOD product was clearly identified and characterized in its functional activity including in vivo action, this manuscript can be considered for publication in this journal, Marine Drugs. However, the following concerns should be addressed or properly revised before acceptance.

1. The Flag tag, which is conventionally used in recombinant protein preparation for antibody-aided detection, is no more recognized as a novel fusion tag, although the authors used it to facilitate the production. As such, title and abstract of this manuscript should be revised to address that the authors used flag tag. The expression of ‘novel’ for flag tag should be restricted throughout the manuscript.

2. It should be addressed why the authors chose yeast as the expression host of the flag-fused AOD. Although AOD is an antibacterial peptide, it might be inert in its flag-fused state. In addition, the recombinant AOD has no bactericidal effect against gram negative bacteria (line 259). Have the authors ever tested whether the flag-AOD can be expressed in E. coli?

3. The English of the manuscript could be further polished including main text and figure legends, although it does not have serious flaws. Spacing, punctuation, and typos (e.g. ‘Lg’ at line 124) should be carefully checked throughout the manuscript.

4. The authors obtained 587 mg/L of total protein from 5 L fermentation. However, the production yields of 6His-2Flag-AOD and final AOD product should be estimated during the downstream process.

5. (Section 2.1) The rationale why two flag tags were fused should be explained. Is one or more than two flag tags not effective?

6. (Figure 1A) Resolution and font size of the figure should be enhanced. Describing all basic components in the pPIC vector plasmid are not necessary. In contrast, restriction sites and full amino acid sequence including all tags of the protein expressed should be depicted.

7. (Figure 1B and 1D) The expected position of the target band should be indicated. Only lane 5 in Figure 1D shows the band corresponding to 6His-2Flag-AOD. However, why is the band not detected in the other lanes in Figures 1B and 1B? As the protein has Flag tag, it could be possible to detect the protein by immunoblot.

8. (Figure 1F) Resolution and font size of the figure should be enhanced. Even briefly, experimental procedure of the mass spectrometry should be described in ‘Materials and Methods’ section or in this figure legend.

9. (Section 2.4 and line 259) As the activity was tested against only bacteria, in section 2.4, “antibacterial activity” is more appropriate than “antimicrobial activity”. In the discussion (line 259), the authors stated that the recombinant AOD has no antibacterial effect against gram negative bacteria. This experiment and result should be also contained in “Materials and Methods” and “Results” sections. In addition, Is there any specific reason why the authors used lincomycin rather than other antibiotics as a control?

10. (Section 2.5 and Figure 2A) The data in Figure 2A has no error bars. I cannot find “some little increase in bacterial count at 8 hr (line 129)”. Even though there are increase at 4 and 8 hrs, is it reproducible? I suggest repeated experiments that can providing error bars on the data.

11. (Figures 2~7) What is “CK”? It should be defined in figure legends.

12. (Sections 2.9 and 2.10) These two sections can be joined together as one section.

13. (Section 4.2.3.) The following points should be described. 1) How did the authors prepare enterokinase? 2) The tag-cleavage reaction procedure should be described more in detail including the amount of enterokinase and substrate for the reaction. 3) After cleavage, target protein should be separated from enterokinase and cleaved flag tag in the reaction mixture. How was the final product purified?

Comments on the Quality of English Language

The English of the manuscript could be further polished including main text and figure legends, although it does not have serious flaws. Spacing, punctuations, and typos (e.g. ‘Lg’ at line 124) should be carefully checked throughout the manuscript.

Author Response

Question 1: The Flag tag, which is conventionally used in recombinant protein preparation for antibody-aided detection, is no more recognized as a novel fusion tag, although the authors used it to facilitate the production. As such, title and abstract of this manuscript should be revised to address that the authors used flag tag. The expression of ‘novel’ for flag tag should be restricted throughout the manuscript.

Answer 2: Thank you for your suggestion. The description of "FLAG" has been revised in the title and abstract.

Title: High yield preparation of American Oyster Defensin (AOD) via small and acidic fusion tag and its functional characterization

Abstract: “a novel short and acidic fusion tag” has been revised to “a short and acidic fusion tag”

Question 2:  It should be addressed why the authors chose yeast as the expression host of the flag-fused AOD. Although AOD is an antibacterial peptide, it might be inert in its flag-fused state. In addition, the recombinant AOD has no bactericidal effect against gram negative bacteria (line 259). Have the authors ever tested whether the flag-AOD can be expressed in E. coli?

Answer 2: Thank you for your comments. The E. coli and P. pastoris were initially employed as expression hosts in the preliminary phase of this study. However, there was almost no target protein was detected in the supernatant of the lysate following IPTG induction. Meanwhile, very low yield of target fusion protein (20 mg/L) was found in precipitates containing inclusion bodies and cell fragments. In contrast, P. pastoris exhibited a higher yield for the FLAG-AOD expression, and it was selected as expression host in this article.

Question 3: The English of the manuscript could be further polished including main text and figure legends, although it does not have serious flaws. Spacing, punctuation, and typos (e.g. ‘Lg’ at line 124) should be carefully checked throughout the manuscript.

Answer 3: Thank you for your comments. The grammatical errors and colloquial expressions were revised. Additionally, the language has been improved by the Elsevier Language Editing Services (No. ASLESTD1247326).

Question 4:  The authors obtained 587 mg/L of total protein from 5 L fermentation. However, the production yields of 6His-2Flag-AOD and final AOD product should be estimated during the downstream process.

Answer 4: Thank you for your suggestion. The final yield of AOD (216 mg/L) was added in the abstract and main text.

Question 5: (Section 2.1) The rationale why two flag tags were fused should be explained. Is one or more than two flag tags not effective?

Answer 5:Thank you for your comments. The antimicrobial peptide AOD carries +5 charges, while each flag-tag possess -3 charges. By employing two flag tags, the positive charge of AOD can be effectively neutralised, leading to a more stable fusion protein. In fact, only one flag tag has been used in our previous attempt but no target product was expressed.

Question 6: (Figure 1A) Resolution and font size of the figure should be enhanced. Describing all basic components in the pPIC vector plasmid are not necessary. In contrast, restriction sites and full amino acid sequence including all tags of the protein expressed should be depicted.

Answer 6: Thanks to your suggestion, Figure 1A has been revised to add the restriction sites and full amino acid sequence including all tags of the protein expressed.

Question 7: (Figure 1B and 1D) The expected position of the target band should be indicated. Only lane 5 in Figure 1D shows the band corresponding to 6His-2Flag-AOD. However, why is the band not detected in the other lanes in Figures 1B and 1B? As the protein has Flag tag, it could be possible to detect the protein by immunoblot.

Answer 7: Thank you for your question. The target fusion peptide 6His-2Flag-AOD was detected in the line 1-6 (Fig. 1B) and line 1,2,5 (Fig. 1D). The presence of different buffers in the samples and standard marker may lead to a discrepancy between the electrophoretic bands and the expected molecular weight (line 1-6 in Fig.1B and line 1,2 in Fig. 1D). Meanwhile, the solution buffer of samples in the line 3-5 (Fig. 1D) contained imidazole, which may contribute the differences in bands. Furthermore, the results of MALDI-TOF MS showed that the line 1 and line 5 in Fig. 1D had the same molecular weight of 7073.439 Da, and it was consistent with the theoretical molecules of 7074 Da.

MALDI-TOF MS analysis of the fermentation supernatant (Line 1 in the Fig. 1D in manuscript) 

MALDI-TOF MS analysis of the eluent at target peak which containing 500 mM imidazole (Line 1 in the Fig. 1D in manuscript)

Question 8: (Figure 1F) Resolution and font size of the figure should be enhanced. Even briefly, experimental procedure of the mass spectrometry should be described in ‘Materials and Methods’ section or in this figure legend.

Answer 8: Thank you for your advice. The resolution and font size of the figure has been improved. The methods and experimental processes of mass spectrometry have been appended to the materials and methods section.

Question 9: (Section 2.4 and line 259) As the activity was tested against only bacteria, in section 2.4, “antibacterial activity” is more appropriate than “antimicrobial activity”. In the discussion (line 259), the authors stated that the recombinant AOD has no antibacterial effect against gram negative bacteria. This experiment and result should be also contained in “Materials and Methods” and “Results” sections. In addition, Is there any specific reason why the authors used lincomycin rather than other antibiotics as a control?

Answer 9: Thank you for your suggestion. The article has updated "antimicrobial activity" to "antibacterial activity". Additionally, as recommended, gram-negative bacteria have been added as a control in the material methods and results. Lincomycin was chosen as the control due to its common usage for treating skin diseases, and its affordability compared to mupirocin.

Question 10: (Section 2.5 and Figure 2A) The data in Figure 2A has no error bars. I cannot find “some little increase in bacterial count at 8 hr (line 129)”. Even though there are increase at 4 and 8 hrs, is it reproducible? I suggest repeated experiments that can providing error bars on the data.

Answer 10: Thank you for your suggestion. 1) The 1×MIC treatments showed regrowth in bacterial counts at 2 and 4 hours, and it has been revised in manuscript. 2) The experiment has undergone replication, and the image has been revised to incorporate error bars.

Question 11: (Figures 2~7) What is “CK”? It should be defined in figure legends.

Answer 11: “CK” is the PBS-treated negative control group, and it has been defined in figure legends from Figures 2-7.

Question 12: (Sections 2.9 and 2.10) These two sections can be joined together as one section.

Answer 12: Thank you for your suggestion. The two sections have been merged as you proposed.

Question 13:  (Section 4.2.3.) The following points should be described. 1) How did the authors prepare enterokinase? 2) The tag-cleavage reaction procedure should be described more in detail including the amount of enterokinase and substrate for the reaction. 3) After cleavage, target protein should be separated from enterokinase and cleaved flag tag in the reaction mixture. How was the final product purified?

Answer 13: Thank you for your question. 1) The recombinant enterokinase with a specification of 1000U and product number P4237-1000U was acquired from Biyuntian Co., Ltd. (Beijing China). 2) The cutting reaction was performed adhering to the instructions, using a 25mM Tris-HCl (pH8.0) buffer containing Flag-tag fusion protein at a concentration of 0.1-1mg/ml. EK was reconstituted to 0.1-0.2U and the enzyme digestion was carried out at 25℃ overnight. 3) Following enzyme digestion, the mixture is subjected to dialysis using a 1kD molecular weight dialysis bag to eliminate salt ions and cut fragments. Then, centrifugation is performed using an ultrafiltration tube with a 10kD transmission rate to remove enterokinase, leaving behind the desired fusion protein in the resulting liquid.

Reviewer 3 Report

Comments and Suggestions for Authors

The work is interesting and well written with the exception of the minor details suggested below.

Language and typing:

The whole text reads very easy and no major language issues were found. There are only these minor points that need correction:

- Line 117: “S. hyicus437-2,” suggested to set an space between hyicus and the number as shown on the rest of bacteria strains

- Line 230: “inyster”… is it “in oyster” or maybe “in cluster”? suggested a correction

Content:

As mentioned, the content is very complete and well presented. The major issues found that need to be clarified or corrected are these:

- Line 100-101 figure 1. All figures A to F are well explained. Simply suggest to rearrange them so there is a logic order to appear. For instance figures B, C, D and E follow a logical order but A is next to F and does not. This is a very picky point to bring up but just a suggestion to name the figures in that logical order.

- Lines 114-120 and table 1 show two apparently, discrepancies in the strain numbers for some bacteria. For instance S. aureus is numbered CvCC 546 in line 16 but the three S. aureus on table 1 do not have the same number. This is also true for S. hyicus 437-2 (line 117) and S. hyicus NCTC 10350 on table 1. If this is done on purpose, it needs explanation. If not, a correction. I made a search in the whole text and I could not find the explanation.

- Line 192: “The infected mice exhibited a tendency to aggregate into clusters within one to four 192 days post-infection…” please, clarify what aggregates. I do not think the mice aggregate. Maybe the bacteria causing infection?

Comments on the Quality of English Language

Please, see comments for authors

Author Response

Response to the comment of Reviewer 3#

Question 1: Line 117: “S. hyicus437-2,” suggested to set an space between hyicus and the number as shown on the rest of bacteria strains

Answer 1: Thank you for your suggestion. A typo has been added between hyicus and the number.

Question 2: Line 230: “inyster”… is it “in oyster” or maybe “in cluster”? suggested a correction

Answer 2: The mistake has been rectified; it should state "in oyster" and has been revised.

Question 3: Line 100-101 figure 1. All figures A to F are well explained. Simply suggest to rearrange them so there is a logic order to appear. For instance figures B, C, D and E follow a logical order but A is next to F and does not. This is a very picky point to bring up but just a suggestion to name the figures in that logical order.

Answer 3: Thank you for your suggestion. We have adjusted the order of the pictures.

Question 4: Lines 114-120 and table 1 show two apparently, discrepancies in the strain numbers for some bacteria. For instance S. aureus is numbered CvCC 546 in line 16 but the three S. aureus on table 1 do not have the same number. This is also true for S. hyicus 437-2 (line 117) and S. hyicus NCTC 10350 on table 1. If this is done on purpose, it needs explanation. If not, a correction. I made a search in the whole text and I could not find the explanation.

Answer 4: Thank you for pointing out the error in the article, it has been corrected.

Question 5: Line 192: “The infected mice exhibited a tendency to aggregate into clusters within one to four 192 days post-infection…” please, clarify what aggregates. I do not think the mice aggregate. Maybe the bacteria causing infection?

Answer 5: Thank you for your comments. The phenomenon of rats congregating has been observed in various experiments. Typically, normal mice within their cages exhibit little physical proximity, remain in good mental health, and only cluster whilst sleeping. However, in the first three days post-infection, observed in this experiment, the mice were less active, lacking energy, and formed congregations. This could be attributed to a bacterial infection.

Round 2

Reviewer 1 Report

Comments and Suggestions for Authors

The Authors answered to most of the comments of the referee concerning the first version of the manuscript except that concerning the presence or not of a reducing agent in the SDS PAGE samples.

In addition, the band present in lane 5 of Fig. 1D seems to have the same mobility of the band present in lane 2 of Fig. 1E (i.e., the final product after cleavage with enterokinase), when it should have the same mobility of the band in lane 1 of Fig. 1E (i.e., the fusion product). The explanation given by the Authors is not convincing (see below comment to Lines 93-94). 

The revised version produced by the Authors still contains inaccuracies. A list is given below:

Line 16: … derived from Crassostrea virginica exhibits a potent … instead of … derived from Crassostrea virginica, and exhibits …. 

Line 17: charge not charges

Line 17: tags not tag

Line 26: … against the clinical strain S. epidermidis G-81…

Line 27: … activity of AOD was unchanged ….

Line 37: peptides not peptide

Line 47: skip … on account of its high charge and hydrophobicity … This is a hypothesis of the Authors of the present manuscript not present in refs. 6 and 7

Lines 51-52: … include His-tag [9], SUMO-tag [10] and GST-tag.

Line 56: short instead of concise

Line 69 skip “for the first time”. It is not the first time that such fusion was used to construct expression vectors (see ref. 11 of the original version that has been removed from the last version. This reference - Krachmarova, E.; Tileva, M.; Lilkova, E.; Petkov, P.; Maskos, K.; Ilieva, N.; Ivanov, I.; Litov, L.; Nacheva, G. His-FLAG 511 Tag as a Fusion Partner of Glycosylated Human Interferon-Gamma and Its Mutant: Gain or Loss? BioMed Research Internation-512 al 2017, 2017 - should be reintroduced in addition to ref 11 - Hopp et al. – of the last version.

Line 90: what does it mean “penetration peak”? Is the peak containing the material not retained by the column?

Lines 93-94: Skip the whole sentence highlighted in yellow “The variation in imidazole content … seen in distinct SDS-PAGE diagrams.“ In lines 357-8 it is said that the material was dialysed before Tricine SDS-PAGE analyses, so the imidazole was removed and cannot interfere in the SDS-PAGE.

Line 95: “The enterokinase cleaved and purified product is shown in Fig. 1E” instead of the sentence reported in the manuscript.

Legend to Fig. 1, line 102: Expression, purification and cleavage of ….

Legend to Fig 1, line 109: Lane 2, unpurified fermentation supernatant after filtration with a 0.45 mM filter

Legend to Fig. 1, lines 109-110: Lane 3, material unretained by the column (if I have understood correctly)

Legend to Fig. 1, lines 111: (E) AOD after cleavage with enterokinase

Line 116: S. aureus ATCC 25923

Line 117: pseudintermedius

Table 1: Salmonella typhimurium ATCC 14028

Table 1: Shigella flexneri: what is CMCC?

After line 120 in paragraph 2.4 add a comment on the lack of activity against the three gram-negative species that have been added in Table 1

Line 122: -: not tested

Line 138: the cell survival rate at 128 mg/ml of AOD

Fig 2D: what is Pepase? Is it pepsin?

Fig 2G: Are the ions concentrations those reported in lines 398 – 399? Were the different ions tested alone or all together in the same solution?  

Line 158: ions not ion

Line 159: (flow cytometry) instead of (FACS)

Line 168: on cell membrane of S. epidermidis

Line 177: skip and

Line 188: Transmission electron microscopy analysis of the effect on S. epidermidis of AOD at 4xMIC after treatment for 1 h. 

Line 190: of instead of for

Lines 195, 197, 206 and 210: treated instead of treatment

Line 211: Compared to the treated group, 

Fig. 6: in the fourth raw CK instead of S. epidermidis G-91

Line 249: neutralized instead of negated 

Line 250: skip P. pastoris

Line 260: hydrolysis with enterokinase instead of enterokinase dissection

Line 261: the final yield of 216 mg/L

Line 266: does S. suis correspond to S. hyicus?

Line 273: have instead of has

Line 279 AOD showed a hemolysis rate

Line 284: The presence instead of the existence

Line 292: The bactericidal mechanism of most of the antimicrobial peptides is directly on bacterial cell membranes

Line 219: treated group

Line 306: in the biomaterial-associated infection model (BAI)

Line 308: In a previous study we set up a murine abscess model for S. epidermidis …

Line 321: was a present of Professor ….

Line 335: T4 DNA ligase was employed

Line 338: plates instead of plate

Line 348: was instead of were

Line 353: in a gradient manner: explain which gradient was used

Line 355: The purified product instead of The the purified product

Line 356: for 16 h with 0.1 – 0.2 U enterokinase

Line 370: each experiment instead of each gradient

Line 378: and subjected to 10-fold serial dilutions followed

Line 395: to evaluate the stability in the presence of proteases

Line 418: and then transferred to fresh medium to reach the logarithmic phase of growth

Line 469: displays instead of display

Comments on the Quality of English Language

Some improvements are needed

Author Response

Question 1: The Authors answered to most of the comments of the referee concerning the first version of the manuscript except that concerning the presence or not of a reducing agent in the SDS PAGE samples.

In addition, the band present in lane 5 of Fig. 1D seems to have the same mobility of the band present in lane 2 of Fig. 1E (i.e., the final product after cleavage with enterokinase), when it should have the same mobility of the band in lane 1 of Fig. 1E (i.e., the fusion product). The explanation given by the Authors is not convincing (see below comment to Lines 93-94). 

Answer 1: The target fusion peptide 6His-2Flag-AOD was detected in the line 1-6 (Fig. 1B), line 1,2,5 (Fig. 1D) and line 1 (Fig. 1E). The presence of different buffers in the samples and standard marker may lead to a discrepancy between the electrophoretic bands and the expected molecular weight (line 1-6 in Fig.1B and line 1,2 in Fig. 1D). Meanwhile, the solution buffer of samples in the line 3-5 (Fig. 1D) contained imidazole, which may contribute the differences in bands. Furthermore, the results of MALDI-TOF MS showed that the line 1 and line 5 in Fig. 1D had the same molecular weight of 7073.439 Da, and it was consistent with the theoretical molecules of 7074 Da.

MALDI-TOF MS analysis of the fermentation supernatant (Line 1 in the Fig. 1D in manuscript)

MALDI-TOF MS analysis of the eluent at target peak which containing 500 mM imidazole (Line 1 in the Fig. 1D in manuscript)

The revised version produced by the Authors still contains inaccuracies. A list is given below:

Question 2: Line 16: … derived from Crassostrea virginica exhibits a potent … instead of … derived from Crassostrea virginica, and exhibits …. 

Answer 2: Thank you for your suggestion. It has been revised.

Question 3 : Line 17: charge not charges

Answer 3: Thank you for your suggestion. It has been revised.

Question 4: Line 17: tags not tag

Answer 4: Thank you for your suggestion. It has been revised.

Question 5: Line 26: … against the clinical strain S. epidermidis G-81…

Answer 5: Thank you for your suggestion. It has been revised.

Question 6: Line 27: … activity of AOD was unchanged ….

Answer 6: Thank you for your suggestion. It has been revised.

Question 7: Line 37: peptides not peptide

Answer 7: Thank you for your suggestion. It has been revised.

Question 8: Line 47: skip … on account of its high charge and hydrophobicity … This is a hypothesis of the Authors of the present manuscript not present in refs. 6 and 7

Answer 8: Thank you for your suggestion. It has been revised.

Question 9: Lines 51-52: … include His-tag [9], SUMO-tag [10] and GST-tag.

Answer 9: Thank you for your suggestion. It has been revised.

Question 10: Line 56: short instead of concise

Answer 10: Thank you for your suggestion. It has been revised.

Question11: Line 69 skip “for the first time”. It is not the first time that such fusion was used to construct expression vectors (see ref. 11 of the original version that has been removed from the last version. This reference - Krachmarova, E.; Tileva, M.; Lilkova, E.; Petkov, P.; Maskos, K.; Ilieva, N.; Ivanov, I.; Litov, L.; Nacheva, G. His-FLAG 511 Tag as a Fusion Partner of Glycosylated Human Interferon-Gamma and Its Mutant: Gain or Loss? BioMed Research Internation-512 al 2017, 2017 - should be reintroduced in addition to ref 11 - Hopp et al. – of the last version.

Answer 11: Thank you for your comment and advice. It has been deleted.

Question 12: Line 90: what does it mean “penetration peak”? Is the peak containing the material not retained by the column?

Answer 12: Penetration peak is material unretained by the column.

Question 13: Lines 93-94: Skip the whole sentence highlighted in yellow “The variation in imidazole content … seen in distinct SDS-PAGE diagrams.“ In lines 357-8 it is said that the material was dialysed before Tricine SDS-PAGE analyses, so the imidazole was removed and cannot interfere in the SDS-PAGE.

Answer 13: Thank you for your comments. The line 3-5 in Figure 1D showed the detection of collected samples during the purification process, and they all contained imidazole.

Question 14: Line 95: “The enterokinase cleaved and purified product is shown in Fig. 1E” instead of the sentence reported in the manuscript.

Answer 14: Thank you for your suggestion. It has been revised.

Question15: Legend to Fig. 1, line 102: Expression, purification and cleavage of ….

Answer 15: Thank you for your suggestion. It has been revised.

Question16: Legend to Fig 1, line 109: Lane 2, unpurified fermentation supernatant after filtration with a 0.45 mM filter

Answer 16: Thank you for your suggestion. It has been revised.

Question17: Legend to Fig. 1, lines 109-110: Lane 3, material unretained by the column (if I have understood correctly)

Answer 17: Thank you for your suggestion. It has been revised.

Question18: Legend to Fig. 1, lines 111: (E) AOD after cleavage with enterokinase

Answer 18: Thank you for your suggestion. It has been revised.

Question19: Line 116: S. aureus ATCC 25923-

Answer 19: Thank you for your suggestion. It has been revised.

Question20: Line 117: pseudintermedius

Answer 20: Thank you for your suggestion. It has been revised.

Question21: Table 1: Salmonella typhimurium ATCC 14028

Answer 21: Thank you for your suggestion. It has been checked and the CVCC is the National Center for Veterinary Culture Collection in China.

Question22: Table 1: Shigella flexneri: what is CMCC?

Answer 22: Thank you for your question. CMCC is the National Center for Medical Culture Collections in China.

Question23: After line 120 in paragraph 2.4 add a comment on the lack of activity against the three gram-negative species that have been added in Table 1

Answer 23: Thank you for your suggestion. Comments on the inactivity of the three Gram-negative bacteria have been added at the corresponding places in the paper.

Question24: Line 122: -: not tested

Answer 24: Thank you for your suggestion. It has been revised.

Question25: Line 138: the cell survival rate at 128 mg/ml of AOD

Answer 25: Thank you for your suggestion. It has been revised.

Question26: Fig 2D: what is Pepase? Is it pepsin?

Answer 26: Thank you for bringing this error. The correct one is pepsin. It's been revised.

Question27: Fig 2G: Are the ions concentrations those reported in lines 398 – 399? Were the different ions tested alone or all together in the same solution?  

Answer 27: Thank you for your question. The ions concentrations are those in lines 398-399 rows; each ion being individually measured for stability.

Question28: Line 158: ions not ion

Answer 28: Thank you for your suggestion. It has been revised.

Question29: Line 159: (flow cytometry) instead of (FACS)

Answer 29: Thank you for your suggestion. It has been revised.

Question30: Line 168: on cell membrane of S. epidermidis

Answer 30: Thank you for your suggestion. It has been revised.

Question31: Line 177: skip and

Answer 31: Thank you for your suggestion. It has been revised.

Question32: Line 188: Transmission electron microscopy analysis of the effect on S. epidermidis of AOD at 4xMIC after treatment for 1 h. 

Answer 32: Thank you for your suggestion. It has been revised.

Question33: Line 190: of instead of for

Answer 33: Thank you for your comments. It has been confirmed.

Question34: Lines 195, 197, 206 and 210: treated instead of treatment

Answer 34: Thank you for your comments. It has been revised.

Question35: Line 211: Compared to the treated group

Answer 35: Thank you for your comments. It has been revised.

Question36: Fig. 6: in the fourth raw CK instead of S. epidermidis G-91

Answer 36: Thank you for your comments. It has been revised.

Question37: Line 249: neutralized instead of negated 

Answer 37: Thank you for your comments. It has been revised.

Qustion38: Line 250: skip P. pastoris

Answer 38: Thank you for your comments. It has been revised.

Question39: Line 260: enterokinase dissection instead of enterokinase dissection

Answer 39: Thank you for your comments. It has been revised.

Question40: Line 261: the final yield of 216 mg/L

Answer 40: Thank you for your comments. It has been revised.

Question41: Line 266: does S. suis correspond to S. hyicus?

Answer 41: Thank you for bringing this error. The correct one is  S. hyicus. It's been modified.

Question42: Line 273: have instead of has

Answer 42: Thank you for your comments. It has been revised.

Question43: Line 279 AOD showed a hemolysis rate

Answer 43: Thank you for your comments. It has been revised.

Question44: Line 284: The presence instead of the existence

Answer 44: Thank you for your comments. It has been revised.

Question45: Line 292: The bactericidal mechanism of most of the antimicrobial peptides is directly on bacterial cell membranes

Answer45: Thank you for your comments. It has been revised.

Question46: Line 219: treated group

Answer 46: Thank you for your comments. It has been revised.

Question47: Line 306: in the biomaterial-associated infection model (BAI)

Answer 47: Thank you for your comments. It has been revised.

Question48: Line 308: In a previous study we set up a murine abscess model for S. epidermidis …

Answer 48: Thank you for your comments. It has been revised.

Question49: Line 321: was a present of Professor ….

Answer 49: Thank you for your comments. It has been revised.

Question50: Line 335: T4 DNA ligase was employed

Answer 50: Thank you for your comments. It has been revised.

Question51: Line 338: plates instead of plate

Answer 51: Thank you for your comments. It has been revised.

Question52: Line 348: was instead of were

Answer 52: Thank you for your comments. It has been revised.

Question53: Line 353: in a gradient manner: explain which gradient was used

Answer 53: Thank you for your comments. It has been revised.

Question54: Line 355: The purified product instead of The the purified product

Answer 54: Thank you for your comments. It has been revised.

Question55: Line 356: for 16 h with 0.1 – 0.2 U enterokinase

Answer 55: Thank you for your comments. It has been revised.

Question56: Line 370: each experiment instead of each gradient

Answer 56: Thank you for your comments. It has been revised.

Question57: Line 378: and subjected to 10-fold serial dilutions followed

Answer 57: Thank you for your comments. It has been revised.

Question58: Line 395: to evaluate the stability in the presence of proteases

Answer 58: Thank you for your comments. It has been revised.

Question59: Line 418: and then transferred to fresh medium to reach the logarithmic phase of growth

Answer 59: Thank you for your comments. It has been revised.

Question60: Line 469: displays instead of display

Answer 60: Thank you for your comments. It has been revised.

Reviewer 2 Report

Comments and Suggestions for Authors

The authors responded to all my comments. However, some points were not reflected in the revised manuscripts. I suggest the following revisions before final acceptance.

1. (Title) Please consider changing “~ via small ~” to “~ via a small ~”.

2. The contents in Answer 2 should be contained in the manuscript (in Results and/or Discussion sections). This would redound to the advantage of using P. pastoris.

3. The contents in Answer 5 should be contained in the manuscript (in Results and/or Discussion sections).

4. (Answer 7) The target bands should be indicated in Figure 1D too. The discrepancy in target band positions should be explained in the manuscript. In particular, lane 5, which contained uncut 6His-2Flag-AOD, shows different position of target band.

5. The contents in Answer 13 should be contained in the manuscript (in Materials and Methods).

6. Throughout the manuscript, “PBS buffer” should be change to just “PBS”.

7. Typo at line 240 (“~ effectively n the ~”) should be corrected.

Comments on the Quality of English Language

Throughout the manuscript, “PBS buffer” should be change to just “PBS”.

Typo at line 240 (“~ effectively n the ~”) should be corrected.

If the manuscript is further revised, the revised parts should be checked in English language quality.

Author Response

Response to Reviewer 2:

Question 1: (Title) Please consider changing “~ via small ~” to “~ via a small ~”.

Answer 1: Thank you for your advice. We have revised the title according to your suggestion.

Question 2: The contents in Answer 2 should be contained in the manuscript (in Results and/or Discussion sections). This would redound to the advantage of using P. pastoris.

Answer 2: Thank you for your suggestion. We've already added that to the discussion section as following:

Initially, E. coli and P. pastoris were employed as expression hosts, but the superna-tant of the lysate following IPTG induction had almost no target protein detected. Addi-tionally, a very low yield of target fusion protein (20 mg/L) was found in the precipitates, which contained inclusion bodies and cell fragments. In contrast, P. pastoris demonstrated a higher yield for the Flag-AOD expression and was subsequently chosen as the expression host.

Question 3: The contents in Answer 5 should be contained in the manuscript (in Results and/or Discussion sections).

Answer 3: Thank you for your comments. We've already added that to the discussion section as following:

The AOD has a net charge of +5, whereas each flag-tag bears a charge of -3. By incorporating two flag tags, the positive charge of AOD can be efficiently negated, resulting in a fusion protein that is more stable.

Question 4: (Answer 7) The target bands should be indicated in Figure 1D too. The discrepancy in target band positions should be explained in the manuscript. In particular, lane 5, which contained uncut 6His-2Flag-AOD, shows different position of target band.

Answer 4: Thank you for your suggestion. We have made modifications to the Fig.1D and included an analysis in the results section as following:

The variation in imidazole content could be the possible cause of the 6His-2Flag-AOD's diverse migration degrees seen in distinct SDS-PAGE diagrams.

Question 5: The contents in Answer 13 should be contained in the manuscript (in Materials and Methods).

Answer 5: Thank you for your advice. We've already added that to the Materials and Methods section as following:

The the purified product of 6His-2Flag-AOD (0.1-1 mg/mL) was enzymatically cleaved in the 25mM Tris-HCl buffer at 25℃ for 16 h with the 0.1-0.2 U enterokinase.

Question 6: Throughout the manuscript, “PBS buffer” should be change to just “PBS”.

Answer 6: Thank you for your comments. Relevant parts of the article have been modified.

Question 7: Typo at line 240 (“~ effectively n the ~”) should be corrected.

Answer 7: Thank you for your comments. The error has been corrected.

Round 3

Reviewer 2 Report

Comments and Suggestions for Authors

The authors properly responded to my comments and revised the manuscript.

Author Response

My sincerest thanks to you for reviewing.